# GENERALISING MULTI-AGENT COOPERATION THROUGH TASK-AGNOSTIC COMMUNICATION

## ABSTRACT

In cooperative *multi-agent reinforcement learning (MARL)*, existing communication methods are almost exclusively task-specific, necessitating the training of new communication strategies for each unique task. This paper addresses this inherent inefficiency by introducing a task-agnostic, environment-specific communication strategy applicable to any task within a given environment. We pretrain the communication strategy *without* task-specific reward guidance in a self-supervised manner using a set autoencoder. Our objective is to learn a latent Markov state from a *set* of local observations, coming from a variable number of agents. Under mild assumptions, we prove that policies using our latent representations are guaranteed to converge, and upper bound the value error introduced by our Markov state approximation. Our method enables seamless adaptation to novel tasks without relearning or fine-tuning the communication strategy, gracefully supports scaling to more agents than present during training, and detects out-of-distribution events in an environment. Empirical results on diverse MARL scenarios validate the effectiveness of our approach, surpassing task-specific communication strategies in unseen tasks.

## 1 INTRODUCTION

**Motivation.** MARL exacerbates the brittleness of reinforcement learning with non-stationarity and convergence issues (Marinescu et al., 2017; Zhang et al., 2019, Section 3.2). It is often impossible to predict the underlying multi-agent Markov state given only the local observation of an agent. This is due to the inherent partial observability of multi-agent problems as an agent typically has no knowledge of what other agents see. Since we require complete cognisance of the Markov state to solve an MDP, without it, MARL seldom leads to ideal or near-ideal solutions. This partial observability further worsens the limited sample efficiency suffered by single-agent RL (Buckman et al., 2018; Yu, 2018). To alleviate these issues in collaborative settings, many approaches utilise communication to share information between agents (Foerster et al., 2016; Sukhbaatar et al., 2016; Das et al., 2019; Bettini et al., 2023). These methods typically use a differentiable strategy, optimising messages with respect to the reinforcement learning objective.

However, thus far, differentiable communication for cooperative multi-agent learning has been entirely task-driven. Previous works have learned communication strategies for solving riddles (Foerster et al., 2016), traffic control (Sukhbaatar et al., 2016; Das et al., 2019), navigation (Das et al., 2019; Li et al., 2020), and tasks requiring heterogeneous behaviour (Bettini et al., 2023). In every example, agents learn a targeted communication strategy for each task that they are expected to solve. Learning such *task-specific* communication strategies is wasteful and inefficient—particularly given the poor sample efficiency of MARL.

We propose reducing the inefficiency of task-specific communication strategies by learning a *task-agnostic* and *environment-specific* communication strategy. In task-specific strategies, even if the environment does not change, each time that agents learn a new task, they also need to learn a new communication strategy. In contrast, a task-agnostic strategy can be shared for all tasks *within this environment*, erasing the requirement of learning new strategies for each new task. This enables learning specialised cooperation skills for diverse tasks in the environment.

---

Our source code is provided in the supplementary materials, with instructions to reproduce all experiments.

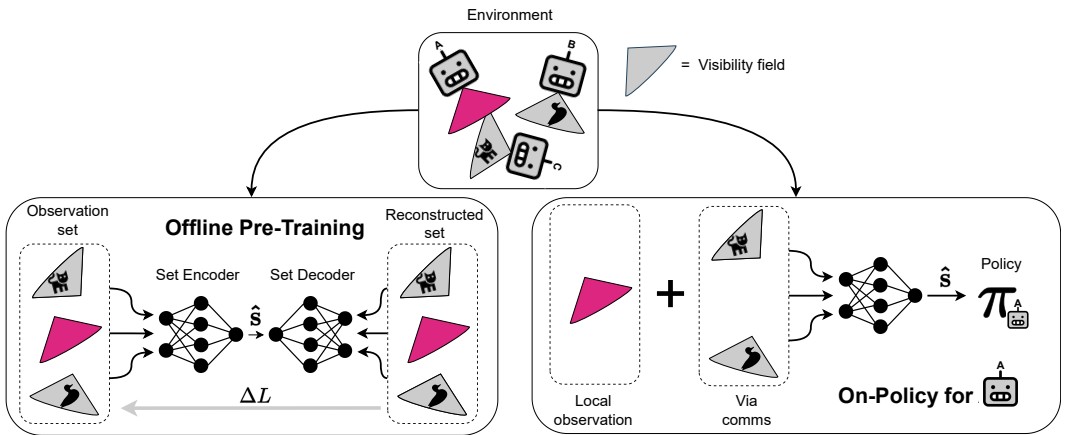

Figure 1: **Learning and applying a task-agnostic communication strategy in MARL.** *Offline pre-training.* We pre-train a set autoencoder with sets of observations collected from exploring an environment. Since there is no reward signal involved in sampling these observations, the autoencoder learns an environment-specific but task-agnostic representation. When a variable number of agent observations are encoded by the set encoder, the output is a fixed-size latent vector ŝ approximating the Markovian state $s$ in a Dec-MDP. *Policy training.* For each agent, we deploy the set encoder on-policy to encode the global observation (assembled via communication) into ŝ. We condition the policy on ŝ so that it acts on the Markov state.

**Approach.** We differentiate between a task (an optimisation objective within an environment) and an environment (a world and its transition dynamics). An agent solves a task in an environment by maximising the return. Formally, the difference is that an environment $\mathcal{E}$ is defined by the state transition probabilities $T_{\mathcal{E}}(s' \mid s, a)$ while a task $\tau$ within $\mathcal{E}$ is defined by a reward function $R_{\tau, \mathcal{E}}(s, a)$.

Noting this distinction, we focus on cooperative multi-agent problems known as Dec-MDPs, where the Markov state is jointly fully observable. In these problems, the state is independent of the task specifics, depending only on the environment. Thus, to achieve perfect cooperation in Dec-MDPs, agents must infer the *global observation* (the joint set of all agents' observations) from a collection of local observations (Oliehoek & Amato, 2016). This is a skill that they need to employ for any given task in an environment. Hence, the skill is *transferable* across tasks, implying that it can be learned through a task-agnostic communication strategy. Our main idea is to learn this task-agnostic communication method by pre-trained reconstruction of the global state (Figure 1).

In addition to eliminating the need to learn new communication strategies for novel tasks in an environment, this method brings several other notable advantages. Firstly, by utilising a specialised set autoencoder, we enable decoding a *variable-sized set* using a *fixed-size latent state*. This permits the communication strategy to elegantly support variable numbers of agents and to even scale *out-of-distribution* to more agents than seen during training. Additionally, by comparing pre-training losses to the losses at runtime, it is possible to detect out-of-distribution disturbances in an environment (e.g. adversarial agents and unsafe environment states). Finally, our approach is *grounded* in the environment, resulting in messages which have specific meaning for any task within this environment. The alternative approach—using a fixed set of communication symbols to achieve environment and task-agnostic communication—is un-grounded.

**Contributions.** We develop a method for learning general, task-agnostic and environment-specific communication strategies in multi-agent teams that supports variable numbers of agents. We provide two proofs which demonstrate that (*i*) under mild assumptions, our method guarantees return convergence and (*ii*) when these assumptions are not met, there is an upper bound on the regret. We test our method with experiments on tasks in VMAS (Bettini et al., 2022) and the Melting Pot suite (Agapiou et al., 2023). Our task-agnostic communication strategy outperforms strategies that are reused (i.e. trained with policy losses on one task and deployed on another task in the same environment). Moreover, we provide evidence that performance does not degrade significantly as we scale the number of agents in the system. Lastly, we showcase how our pre-training method can be used to detect out-of-distribution events in the environment.

## 2 PRELIMINARIES

In this paper, we are mainly interested in a subset of cooperative multi-agent problems known as *decentralised Markov decision processes (Dec-MDPs)*. In the Dec-MDP framework, the environment provides a joint observation $o = \{o_1, \ldots, o_n\}$ from which each agent observes its local observation and decides on an action. Upon taking an action, the global reward function $R$ provides a shared reward for each time step, and each agent receives its next local observation (Bernstein et al., 2002). A key feature of Dec-MDPs is that the underlying Markov state must be *uniquely defined* by the *joint set of observations* of all agents (i.e. the global observation) (Oliehoek & Amato, 2016).

Formally, denoting $\Delta(X)$ as the set of probability distributions over the set $X$, a Dec-MDP is defined by a tuple $(n, S, A, T, \mathbb{O}, O, R, \gamma)$ where $n$ is the number of agents, $S$ is the set of states (with initial state $s_0$), $A$ is the set of actions for each agent, $T : S \times A^n \to \Delta(S)$ is the state transition probability function $T(s' \mid s, \mathbf{a})$, $\mathbb{O}$ is the set of joint observations, $O : S \times A^n \to \Delta(\mathbb{O})$ is the observation probability function $O(\mathbf{o} \mid s, \mathbf{a})$, $R : S \times A^n \to \Delta(\mathbb{R})$ is the global reward function $R(s, \mathbf{a})$, $\gamma$ is the discount factor, and the multi-agent Markov state $s$ is unambiguously determined by $\mathbb{O}$.[1]

## 3 LEARNING TASK-AGNOSTIC COMMUNICATION

In a Dec-MDP, agent cooperation depends on the Markovian state of the multi-agent team within an environment. This state is *independent* of task specifics, consisting of the current state of the *environment*. Therefore, we define a communication model which reconstructs this environment-specific information, and show how it can be trained without reward guidance to be completely task-agnostic.

### 3.1 A COMMUNICATION MODEL FOR TASK-AGNOSTIC COOPERATION

As the global observation defines the multi-agent Markov state in a Dec-MDP, we define our communication model as one in which agents reconstruct the multi-agent state by reconstructing the joint set of all agents' observations.

Consider a Dec-MDP defined by the tuple $(n, S, A, T, \mathbb{O}, O, R, \gamma)$ with agents $i \in \mathcal{A}_n = \{1, \ldots, n\}$. Agents have a communication range $\epsilon$ where if the distance $d(i, j)$ between agents $i$ and $j$ is greater than $\epsilon$ then they cannot share information. Thus, we define the neighbourhood of agent $i$ as $\mathcal{N}_i = \{j \in \mathcal{A}_n \mid d(i, j) \leq \epsilon, j \neq i\}$. In each time step $t$, an agent $i$ receives a set which contains the observations of all agents within $i$'s range,

$$\mathbb{O}_t^{\mathcal{N}_i} = \{o_t^j \mid \forall j \in \mathcal{N}_i\}. \tag{1}$$

Let $\mathbb{O}_t$ denote the joint set of all agent observations in time step $t$. With $\mathbb{O}_t^{\mathcal{N}_i}$ and its local observation $o_t^i$, the agent can recover the set of observations of all agents within the communication range of $i$ (including itself) in this time step,

$$\mathbb{O}_t^i = \mathbb{O}_t^{\mathcal{N}_i} \cup \{o_t^i\}. \tag{2}$$

Using an autoencoder, the set $\mathbb{O}_t^i$ is encoded into a task-agnostic latent state $\hat{\mathbf{s}}_t^i$. This latent state is permutation-invariant and is a constrained approximation of the global observation $\mathbb{O}_t = \{o_t^i, \ldots, o_t^n\}$ (and therefore, the Markov state) constructed using the information available in $\mathbb{O}_t^i$ and the knowledge of the autoencoder. The advantage of this state over a concatenation of all observations is that it is fixed in size, supporting variable numbers of agents, makes use of the sample efficiency afforded by a permutation-invariant state, and is an efficient compressed representation.

To use this approximation of the multi-agent state in decision-making, we condition the policy of agent $i$ on this latent state. Let $\pi_{\theta_t}$ denote a policy parameterised by weights $\theta_t$. The probability that agent $i$ takes action $a_t^i$ is given by $\pi_{\theta_t}(a_t^i \mid \hat{\mathbf{s}}_t^i, o_t^i)$. The policy is conditioned on the agent's local observation, even though $o_t^i$ is encoded within $\hat{\mathbf{s}}_t^i$, because $\hat{\mathbf{s}}_t^i$ is permutation-invariant. Without $o_t^i$, the policy cannot determine which agent it is reasoning about.

When the latent state $\hat{\mathbf{s}}_t$ perfectly captures the global observation, the policy is guaranteed to converge to a local optimum in the return:

---

[1]This notation is inspired by Oliehoek & Amato (2016) and Ellis et al. (2022).

**Theorem 3.1.** *A policy gradient method, which conforms to the assumptions in (Sutton et al., 1999, Theorem 3), conditioned on $\hat{\mathbf{s}}_t$ in a Dec-MDP is guaranteed to converge to a local optimum in the return assuming $\hat{\mathbf{s}}_t$ captures $\mathbb{O}_t$ with zero reconstruction error.*

*Proof.* Consider a Dec-MDP defined by the tuple $(n, S, A, T, \mathbb{O}, O, R, \gamma)$. Define a policy $\pi$ parameterised by $\theta$ that maps the global observation $\mathbb{O}$ to a distribution over joint actions $A$. Formally, $\pi_\theta(\mathbf{a}_t \mid \mathbb{O}_t)$ represents the probability of taking joint action $\mathbf{a}_t$ given global observation $\mathbb{O}_t$ and policy parameters $\theta$. The objective is to optimise the policy $\pi$ to maximise the expected return over a trajectory $\tau = (s_1, \mathbf{a}_1, s_2, \mathbf{a}_2, \ldots, s_T, \mathbf{a}_T)$, where $s_t$ is the multi-agent Markov state at time $t$. Assume a policy gradient method, such as REINFORCE (Williams, 1987; 1992), to update the policy parameters $\theta$. This requires estimating the gradient of the expected return with respect to $\theta$ in order to update these parameters.

Note that (*i*) policy gradient methods converge to a locally optimal policy in Markov decision processes (Sutton et al., 1999, Theorem 3), (*ii*) by definition: the joint state $s_t$ in a Dec-MDP is the multi-agent Markov state (Bernstein et al., 2002; Oliehoek & Amato, 2016), and (*iii*) by definition: this state is jointly fully observable in Dec-MDPs (Bernstein et al., 2002; Oliehoek & Amato, 2016).

Then, since (*iii*) implies the global observation $\mathbb{O}_t$ uniquely defines $s_t$ (Oliehoek & Amato, 2016), and by (*ii*) $\mathbb{O}_t$ defines the multi-agent Markov state, since we use a policy gradient method, by (*i*) it is guaranteed to converge to a local optimum as our policy is conditioned on $\mathbb{O}_t$, which is equivalent to the underlying Markov state. When the latent state $\hat{\mathbf{s}}_t$ captures $\mathbb{O}_t$ with zero reconstruction error, this result extends to when the policy is conditioned on $\hat{\mathbf{s}}_t$ instead. $\square$

However, in practice we cannot assume that $\hat{\mathbf{s}}_t$ captures $\mathbb{O}_t$ with no error. To quantify the effect of any error on the return, we can place a bound on the regret: the difference in the expected return achieved if the approximation of the underlying Markov state was perfect.

**Theorem 3.2.** *Suppose the policy in a Dec-MDP and its associated value function are Lipschitz continuous. Then the regret of a policy learned from an approximation $\hat{\mathbf{s}}_t$ of the underlying Markov state $\mathbf{s}_t$ is bounded above and this bound is directly proportional to the reconstruction error.* [2]

*Proof.* Consider an identical setting to that stated in the proof of Theorem 3.1. Additionally, define a value function $V_{\pi_\theta}$ derived from the policy $\pi_\theta$ and let $\epsilon$ be some error in reconstructing the underlying multi-agent Markov state $\mathbf{s}_t$. Thus, $\hat{\mathbf{s}}_t$ can be decomposed into $\mathbf{s}_t + \epsilon$.

Assume that $V_{\pi_\theta}$ is $K$ Lipschitz continuous where $K \in \mathbb{R}$. Since $V_{\pi_\theta}$ is derived from $\pi_\theta$, let us also assume that $\pi_\theta$ is Lipschitz continuous.

Then,

$$|V_{\pi_\theta}(\mathbf{s}_t) - V_{\pi_\theta}(\hat{\mathbf{s}}_t)| \leq K|\mathbf{s}_t - (\hat{\mathbf{s}}_t)| \tag{3}$$

$$|V_{\pi_\theta}(\mathbf{s}_t) - V_{\pi_\theta}(\mathbf{s}_t + \epsilon)| \leq K|\mathbf{s}_t - (\mathbf{s}_t + \epsilon)| \tag{4}$$

$$= K|-\epsilon| \tag{5}$$

$$= K|\epsilon|. \tag{6}$$

Thus, the difference in expected return (regret) between a policy which assumes the underlying Markov state is the true state $s_t$ and one which assumes the underlying Markov state is an approximation $\hat{\mathbf{s}}_t$ due to reconstruction error $\epsilon$ is bounded above by $K|\epsilon|$. Since $K$ is a constant, this bound is directly proportional to the root mean squared error $|\epsilon|$. $\square$

Hence, in a Dec-MDP, the reconstruction error $\epsilon$ is precisely the autoencoder's error in reconstructing $\mathbb{O}_t$ from the latent state $\hat{\mathbf{s}}_t$ as the Markovian state $\mathbf{s}_t$ is uniquely defined by the global observation. In general, Theorem 3.2 can be extended to Dec-POMDPs as a bound on the error in estimating the global observation rather than the underlying Markov state.

---

[2]For this theorem, we treat $s_t$ as a vector $\mathbf{s}_t$ to decompose its approximation into the true state and error.

## 3.2 TRAINING A TASK-AGNOSTIC COMMUNICATION STRATEGY

We posit that we can learn a task-agnostic communication method by pre-training an autoencoder with global observations $\mathbb{O}_t$ from exploration of an environment. If the exploration policy is independent of the reward function, the strategy is task-agnostic. We use either a uniform random policy or uniform random sampling from the observation space. Neither requires knowing a reward function and hence any method learned in this way is task-agnostic.

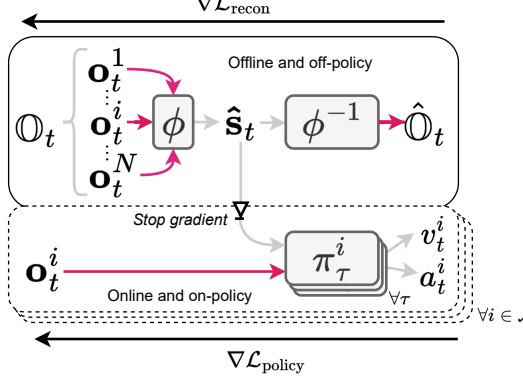

Figure 2: **Method details.** A set autoencoder (with encoder $\phi$ and decoder $\phi^{-1}$) is pre-trained with global observations sampled from reward-free exploration using a self-supervised reconstruction loss offline. During policy training, the pre-trained set autoencoder is loaded with its weights frozen. This communication strategy is shared when training policies for all tasks $\tau$ and agents $i$ within a specific environment. The input to a policy $\pi$ is a concatenation of the encoded set of observations from all agents $\hat{\mathbf{s}}_t$ (an approximation of the Markov state) and the relevant agent's observation $o_t^i$.

Figure 2 provides a detailed overview of our method. Given a permutation-invariant set autoencoder with encoder $\phi$ and decoder $\phi^{-1}$, we train the autoencoder with a self-supervised loss

$$\frac{1}{n} \sum_{t=1}^{n} l\Big(\phi^{-1}(\phi(\mathbb{O}_t)), \mathbb{O}_t\Big), \qquad (7)$$

where $l$ is a function defining a set reconstruction error specified by our choice of autoencoder.

Typically, *graph autoencoders (GAEs)* (Tian et al., 2014; Wang et al., 2016; Kipf & Welling, 2016) would be ideal to encode sets with permutation invariance. However, we instead use the *permutation-invariant set autoencoder (PISA)* (Kortvelesy et al., 2023) because, unlike many GAEs, this architecture allows decoding a variable-sized set using a fixed-size latent state. In other words, no matter the number of agents or the corresponding cardinality of the set $\mathbb{O}_t$, the dimension of the latent state $\hat{\mathbf{s}}_t$ is constant. This property is highly desirable as it allows a trained encoder to scale as agents are added or removed from the environment. If pre-trained on global observations, it also enables the autoencoder to approximate the global

observation even when some observations in the multi-agent team are missing. We also note that to the best of our knowledge, PISA achieves the lowest set reconstruction error among comparable set autoencoders (Kortvelesy et al., 2023, Figure 2), and therefore, is an apt architectural choice for our task-agnostic communication strategy.

While many environments emit two-dimensional pixel observations, PISA encodes feature vectors into permutation-invariant states. Given this dichotomy, when required, we also pre-train a convolutional autoencoder on each element of $\mathbb{O}_t$ to encode each pixel observation $o_t^i$ into a feature vector $\mathbf{v}_t^i$. Thus, when an image encoder is necessary, a set of these feature vectors $\mathbb{V}_t$ is the input to our set autoencoder rather than $\mathbb{O}_t$ directly.

## 4 EXPERIMENTS & DISCUSSION

We propose three experiments. The first shows that a task-agnostic communication strategy is more effective than a task-specific strategy when presented with a novel task. It also verifies that our proposed strategy outperforms a baseline that does not use communication. Our second experiment validates the claim that our method elegantly handles variable numbers of agents. It shows how our method fares as more agents are introduced, going out-of-distribution with respect to the number of agents seen during pre-training. The final experiment demonstrates that, by comparing pre-training autoencoder losses to the losses during policy training, we can detect out-of-distribution events in the environment. Refer to Appendix G for the hardware and implementation details of running these experiments.

## 4.1 EXPERIMENTAL SETUP

Our experiments focus on two MARL suites. Firstly, Melting Pot (Agapiou et al., 2023) is a suite of 2D, grid-based, discrete multi-agent learning environments, providing scenarios that can test a variety of types of coordination focusing on *social dilemmas*. In this type of scenario, problems are a mixture of competition and cooperation so many tasks are not fully cooperative and accordingly, cannot be Dec-MDPs. To address this, we sum all individual agent rewards emitted by the base Melting Pot task into a shared global reward. This ensures each task is fully cooperative. Refer to Appendix A for the complete modification details.

To supplement Melting Pot, we also study tasks in the *vectorised multi-agent simulator (VMAS)* (Bettini et al., 2022): a 2D, continuous-action framework designed for benchmarking MARL. Together, these two suites provide a comprehensive study, as they cover both visual and vector observations, discrete and continuous action spaces, sparse and dense rewards, and different forms of co-operation requirements, ranging from high-level to low-level collaboration strategies. We include further discussion around benchmark selection in Appendix F.

We obtain our pre-training dataset by following a uniform random policy (Melting Pot) or uniform random sampling from the observation space (VMAS) for a million steps in the environment. With these samples, we train a task-agnostic communication strategy (Section 3.2) and deploy it with our communication model (Section 3.1). For all of our experiments, we optimise the policy for each agent using *Proximal Policy Optimisation (PPO)* (Schulman et al., 2017). This is commonly referred to in multi-agent literature as *Independent PPO (IPPO)*. However, we emphasise that our method is algorithm-agnostic. Any optimisation algorithm may be used with our method in place of IPPO. Further details are available in Appendix E (pre-training) and Appendix D (policy training).

## 4.2 PERFORMANCE ON NOVEL TASKS

In this experiment, we measure the converged reward of our method (*task-agnostic*) against two baselines as we learn policies for a variety of tasks. The *task-specific* baseline simulates reusing communication strategies learned from other tasks. In a real use case, this is the only option to avoid training a new strategy if a task-agnostic strategy does not exist. The difference between the task-specific and task-agnostic baselines is in the set autoencoder pre-training. For the task-specific baseline, we pre-train the set autoencoder using reinforcement learning while trying to learn a policy for a distinct but similar task in the same environment. We use an identical setup for pre-training the task-specific set autoencoder as when we evaluate the task-agnostic method. In contrast, the task-agnostic baseline uses random samples from the environment with reconstruction loss for pre-training. This approach lets us assess how well a task-agnostic method generalises compared to a task-specific one using the same architecture. The *no-comms* baseline uses no communication strategy at all. For Melting Pot environments, we additionally utilise an image encoder, pre-trained in all tasks in an environment, which we use for every baseline.

We evaluate our method, along with the baselines, on three distinct tasks (Figure 3):

**Collaborative Cooking: Circuit.** In Melting Pot's collaborative cooking environment, the primary task is for agents to collaborate to complete recipes. In each variant, the task is slightly different as agents must learn different coordination skills to successfully cook together. For our task-agnostic strategy, we pre-train on the environment and deploy it to learn the *Circuit* variant, where two agents must navigate around a circuit to access cooking pots and ingredients. The *task-specific* variant uses a communication strategy which was learned in the *Cramped* variant where agents must cook under tight space restrictions.

Figure 3: **Tasks.** Circuit (top), Discovery (middle), and Pursuit-Evasion (bottom).

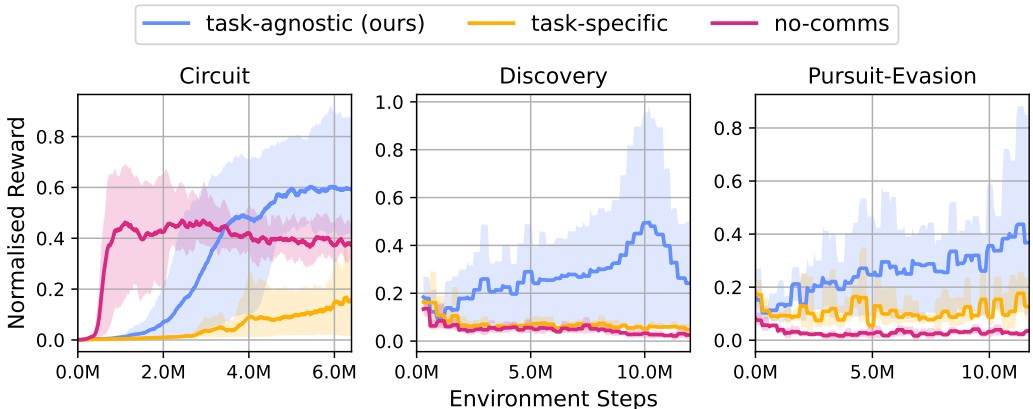

Figure 4: **Task-Agnostic communication strategies lead to greater rewards in novel tasks.** For each set of results, we report the mean and central 95% interval over 5 seeds. We trained for 6.4 million environment steps in Melting Pot tasks, and 12 million environment steps in VMAS tasks. We used the same number of steps to pre-train the task-specific strategy on a similar task.

**Discovery.** In this VMAS task, four agents must try to discover targets. To get a positive global reward, two out of four agents must position themselves within a small radius of a target. This is known as *covering* the target and is how a target is "discovered". Together, agents must coordinate to discover targets as fast as possible. New targets continuously spawn as others are discovered. For this task, the *task-specific* variant uses a communication strategy learned in VMAS' Flocking task where agents must learn to flock, much like birds do, around a moving target.

**Pursuit-Evasion.** The VMAS Pursuit-Evasion task is a find-and-intercept game. The agents are pursuers and they must catch an evading target. The visibility of the pursuers is limited. They must work together to find the target and collectively swarm around them in order to catch them as fast as possible to achieve the highest reward. For this, the *task-specific* variant uses a communication strategy learned in VMAS' Discovery task.

We see a significant improvement in return when using task-agnostic strategies (Figure 4):

In **Circuit**, the task-specific baseline fails to achieve a mean reward much higher than the starting reward. The no-comms baseline sharply rises to a strong reward, but soon plateaus, no longer improving after around 1M steps. In contrast, our task-agnostic method improves more slowly, but after approximately 4M steps of training, outperforms both baselines. This is likely because it takes some time to learn how to decode the permutation-invariant latent state.

In **Discovery**, both the task-specific and no-comms baselines lead to a quick collapse of the reward, followed by plateauing, after approximately 400,000 steps, failing to learn a useful representation. Meanwhile, the task-agnostic strategy produces a better policy almost immediately, gradually improving and peaking after around 10M steps. We outperform the two baselines from just after the start and through to the end of training.

In **Pursuit-Evasion**, while the task-specific baseline appears to outperform the no-comms baseline, much like Discovery, both plateau after a small number of training steps. The task-agnostic communication strategy improves from the very start to the end, outperforming both baselines after only about 1M steps.

The results show that task-agnostic communication strategies consistently enable agents to leverage communication without relearning the communication strategy. This indicates that our approach is a practical alternative to the task-driven methods used in the community. This is useful in the real world, where cooperative robots engage in a variety of tasks in a shared environment. Our method has allowed us to use a general prior (pre-trained) understanding of the environment to generalise to novel tasks better than the baselines. Finally, while using an autoencoder may appear trivial, it is essential to emphasise that our approach is distinct due to being task-agnostic. Other works (Lin et al., 2021; Guan et al., 2022) have also explored autoencoders as a component of representation learning

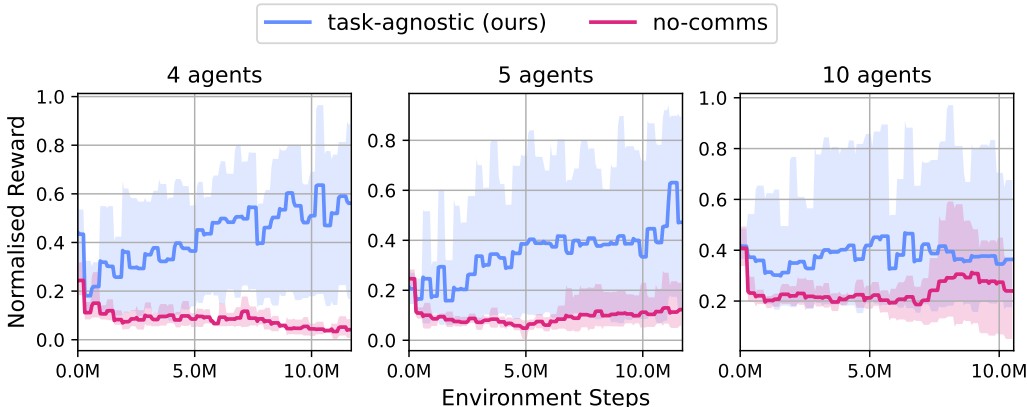

Figure 5: **Our task-agnostic strategy scales out-of-distribution.** We pre-trained the communication strategy with 1, 2, and 3 agents and trained the policy for 12 million environment steps. For each set of results, we report mean and central 95% interval over 5 seeds.

for communication in MARL, but their approaches are decidedly task-specific. Consequently, it is not possible to compare these works to ours.

### 4.3 GENERALISATION WITH OUT-OF-DISTRIBUTION NUMBERS OF AGENTS

In certain real-world situations, new agents may join the multi-agent team during execution to support other agents if help is required, expanding the communication network. To cope with such dynamic scenarios, we investigate how pre-trained reconstructions of the global state generalise to out-of-distribution (OOD) inputs. Since the PISA encoder's latent state is fixed-size, input cardinality is independent of output dimensionality—we can encode a set of any cardinality into a constant size latent vector. Therefore, in this experiment, we measure the performance of our communication strategy when we have more agents, and hence larger sets, than seen during pre-training.

We pre-train our set autoencoder in the Discovery environment with 1, 2, and 3 agents, using 1M samples for each case. Then, we train and evaluate a policy on more agents than seen during pre-training. Under these conditions, in Figure 5, we show that our method still significantly outperforms the baseline when we learn a policy with 4 and 5 agents, going beyond the number of agents that we pre-trained our communication strategy with. This is evidence that our approach can elegantly handle changes in connectivity (e.g. from communication disruptions) and can support variable numbers of agents without fine-tuning. Even once we reach 10 agents, although the performance gap is smaller, we continue to outperform the no-communication baseline. At this point, we are far outside the training distribution.

### 4.4 DETECTING OUT-OF-DISTRIBUTION EVENTS

We often want to detect out-of-distribution events in an environment. For example, if a disturbance occurs that makes it unsafe (e.g. humans entering a robot-only operating area, or adversarial agents accessing the communications network), we want to detect this so that agents can safely halt or take appropriate actions. In this experiment, we show that we can detect these OOD occurrences by comparing the set reconstruction loss during training and at runtime.

In Figure 6, we show how the set reconstruction loss changes when we deploy a communication

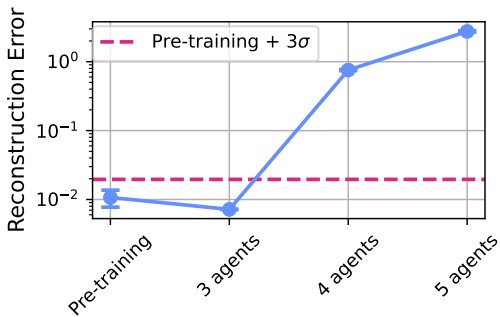

Figure 6: **Detecting OOD agents.** We report the mean set reconstruction error of PISA over the last 10 iterations of policy training.

strategy pre-trained with 1, 2, and 3 agents on Discovery (as in Section 4.3) to train policies with 3 (in-distribution), 4 (OOD), and 5 (OOD) agents in the same environment. A threshold set by the pre-training loss mean plus three standard deviations easily detects the OOD agent counts.

Similarly, we can detect OOD observations. In the Collaborative Cooking environment, when we fix one of the agents to receive only Gaussian noise observations (OOD), the loss exceeds our threshold (Figure 7). Without any observation-tampering (in-distribution), this does not happen.

## 5  RELATED WORK

Addressing MARL's poor sample efficiency is challenging. Prior MARL papers have neglected the inefficiency of relearning communication strategies for distinct tasks. We addressed this by introducing task-agnostic communication strategies that can be shared for all tasks within an environment. Additionally, we support variable numbers of agents—something earlier autoencoder-based communication strategies were unable to do.

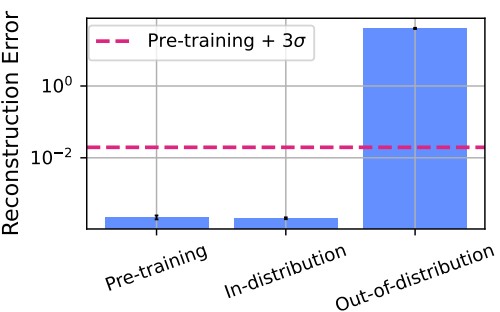

Figure 7: **Detecting OOD observations.** We measure the mean set reconstruction error of PISA over the last 10 iterations of policy training.

**Differentiable Communication.** Differentiable models of communication optimise messages with respect to the objective on-policy. Some typical examples are DIAL (Foerster et al., 2016), CommNet (Sukhbaatar et al., 2016), TarMAC (Das et al., 2019), Eccles et al. (2019), HetGPPO (Bettini et al., 2023), EPC (Long et al., 2020), SPC (Wang et al., 2023), and Abdel-Aziz et al. (2023). All of these methods employ task-specific communication strategies and thus require optimising the strategy for each distinct task. In contrast, our method is task-agnostic—tasks within an environment can share our pre-trained communication strategies. While some of these works support variable numbers of agents (population-invariant communication), DIAL, TarMAC, Eccles et al. (2019) and Abdel-Aziz et al. (2023) do not. Our approach supports population-invariant communication *in addition to* task-agnostic communication through a fixed-size latent state in the autoencoder.

**Self-Supervised Communication.** Several recent works have used self-supervised and contrastive objectives to train differentiable communication strategies (Lin et al., 2021; Guan et al., 2022; Lo & Sengupta, 2022; Lo et al., 2023). All of these methods learn the communications policy *online*, biasing the communications towards a specific objective, while we learn it offline without any bias. Hence, they are not task-agnostic strategies. Furthermore, none of these methods support variable numbers of agents as ours does.

**Pre-training in RL.** Contemporary works in RL have utilised pre-training to leverage prior knowledge when training policies (Cruz et al., 2017; Singh et al., 2021; Yang & Nachum, 2021; Schwarzer et al., 2021; Seo et al., 2022). Fundamentally, they all attempt to learn representations that are useful for solving the underlying MLP through various unsupervised methods. While all of these works focus on single-agent RL, we utilise pre-training to improve the efficiency of MARL.

Other notable related work includes natural language communication, such as Eloff & Engelbrecht (2021), where messages are passed through a restricted discrete set of words. Unlike our approach, this is not continuously differentiable and, due to the selection of communication symbols and RL-based communication training, is not task-agnostic.

## 6  CONCLUSION

We proposed task-agnostic communication strategies to eliminate the inefficiency of task-specific communication, using a set autoencoder to reconstruct the global state from local observations. Our

approach is guaranteed to converge under modest assumptions, with an upper bound on regret due to approximating the Markovian state. Empirically, it outperforms task-specific strategies in novel tasks, scales to more agents than in pre-training, and detects out-of-distribution events during policy training using pre-training losses.

**Limitations.** For simplicity, we use full connectivity between agents. However, this is not a technical limitation since it can be overcome by propagating information via aggregation (e.g. aggregating sets of PISA encodings with another PISA). Additionally, collecting pre-training samples with a scheme such as curiosity-driven exploration (Pathak et al., 2017) could lead to more efficient representations of the Markovian state from the autoencoder as it samples sparse states more frequently.

Nonetheless, as it stands, our method is adaptable to various learning paradigms, not just RL, because it pre-trains communication strategies in a self-supervised manner. As we avoid end-to-end training, we also expedite RL policy training by tuning fewer weights. Additionally, having pre-trained an autoencoder, our policy can use sparse reward signals more efficiently as it does not need to learn environment-specific features. Lastly, our method opens up new applications, allowing changing policies at runtime, or running heterogeneous policies on different collaborative agents.

## REPRODUCIBILITY STATEMENT

Our complete source code is provided in the supplementary materials, including instructions to reproduce every experiment. In addition, we provide extensive details of our network architectures (Appendix C), training and pre-training hyperparameters (Appendix D and E), and compute resources (Appendix G).

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

## A  MODIFICATIONS TO TASKS AND ENVIRONMENTS

As alluded to in Section 4.1, we enforce that Melting Pot tasks are Dec-MDPs with a global reward $R$. In the default case, an agent $i$ receives an individual reward $r_i$. We modify this such that each individual agent reward $r_i$ is summed to make a global reward $R = \sum_i^n r_i$ and this global reward is shared between all agents instead of them receiving their individual reward.

In all collaborative cooking task variants, we enable a pseudo-reward that rewards agents with $+1$ for picking up a tomato and putting it in a pot. Without this pseudo-reward, these tasks have extremely sparse rewards that make them highly challenging to learn. This shaped reward assists in learning on these typically very difficult tasks.

In VMAS, we modify the observation spaces of Discovery and Flocking to be identical by replacing Discovery's redundant additional position vector feature with a two-dimensional zero vector and reducing the number of LiDAR rays from 15 to 12. This allows us to use communication strategies learned in Flocking on Discovery. We argue that Flocking and Discovery are two tasks in the same environment because the obstacles in Flocking can be treated as targets to derive the Discovery task from Flocking (along with a change of reward).

Furthermore, we designed and created Pursuit-Evasion ourselves based on both Flocking and Discovery. This is another task that we argue is in the same environment. It can be derived from Flocking if we treat Flocking's target as a robber, and the agents as the police after modifying the reward. By extension, it can also be derived from Discovery.

## B  PERMUTATION-INVARIANT SET AUTOENCODER ARCHITECTURE

The set autoencoder architecture is based on Kortvelesy et al. (2023, Figure 1). It encodes a variable-sized set of elements $\{x_1, x_2, \ldots, x_n\}$ where $x_i \in \mathbb{R}^n$ into a fixed-size permutation-invariant latent state $z \in \mathbb{R}^z$. It is trained with a self-supervised reconstruction objective to decode the latent state and recover the set.

**Encoder.** The encoder takes the input set $\{x_1, x_2, \ldots, x_n\}$ and maps each element to a key $k_i$ according to some criterion and encodes the keys using a network $\psi_{\text{key}}$. Simultaneously, the encoder takes the input set elements and also encodes them into values using a separate network $\psi_{\text{val}}$. The encoder then takes the element-wise product of the corresponding key and value embeddings and sums them all. Finally, a cardinality embedding $\lambda_{\text{enc}}(n)$ is added to this sum to form the final latent state $z$.

**Decoder.** The decoder takes the latent state $z$ and predicts the cardinality of the set with a network $\lambda_{\text{dec}}$. The predicted cardinality is used to create a set of keys as in the encoder and the keys are mapped to queries by a network $\phi_{\text{key}}$. Each query is element-wise multiplied by the latent state and a final decoder network $\phi_{\text{dec}}$ recovers the set from these embeddings.

Further details may be found in Kortvelesy et al. (2023). The hyperparameters used in our work are detailed in Appendix E.

## C  POLICY NETWORK ARCHITECTURES

When training to solve Melting Pot tasks, we independently train the policy of each agent with PPO. For each agent, we have independent three-layer MLPs as our policy and value networks. The policy network's hidden layer is 128 neurons wide, while the value network's hidden layer is 1024 neurons wide. We initialise the last layer of the policy network and value network using a normal distribution with zero mean and 0.01 standard deviation in line with the suggestions made by Andrychowicz et al. (2020).

Unlike Melting Pot, as no heterogeneous behaviour is required for our VMAS tasks, we train a policy that's shared between all agents with PPO. For each agent, we have independent three-layer MLPs as our policy and value networks. For Discovery, the policy and value networks have a 256-wide hidden layer while for Pursuit-Evasion, the hidden layers are 512-wide. We initialise the last layer of the policy and value networks with the same normal distribution as we use for Melting Pot.

## D    TRAINING HYPERPARAMETERS

Our training hyperparameters are dependent on the multi-agent suite and task and are described in Table 1 and 2. We always use fixed seeds 0-4 for every experiment. Specifically for Pursuit-Evasion, we use the defaults for the parameters in Table 2 except for train batch size, SGD minibatch size, training iterations, and rollout fragment length.

Table 1: Melting Pot training hyperparameters.

| Parameter | Value |
|---|---|
| Train batch size | 6400 |
| SGD minibatch size | 128 |
| Training iterations | 1000 |
| Rollout fragment length | 100 |

Table 2: VMAS training hyperparameters.

| Parameter | Value |
|---|---|
| Train batch size | 60000 |
| SGD minibatch size | 4096 |
| Training iterations | 200 |
| Rollout fragment length | 125 |
| KL coefficient | 0.01 |
| KL target | 0.01 |
| $\lambda$ | 0.9 |
| Clip | 0.2 |
| Value function loss coefficient | 1 |
| Value function clip | $\infty$ |
| Entropy coefficient | 0 |
| $\eta$ | 5e-5 |
| $\gamma$ | 0.99 |

## E    PRE-TRAINING HYPERPARAMETERS

For Melting Pot environments we train an image encoder in addition to PISA. Observations are first encoded with the image encoder before this embedding is passed to PISA. For the image encoder, we use a 3-layer CNN encoder and decoder as specified in Table 3. Our training data is gathered from observations of all agents generated with a uniform random policy rolled out over 1M environment steps. We train the image encoder with a mini-batch size of 32 for approximately 1000 iterations or until the loss has clearly converged.

Table 3: Melting Pot image autoencoder architecture.

| Layer | Type | In ch. | Out ch. | Kernel | Stride | Padding | Activation |
|---|---|---|---|---|---|---|---|
| Encoder | Conv2D | 3 | 16 | 3 | 2 | 1 | ReLU |
|  | Conv2D | 16 | 32 | 3 | 2 | 1 | ReLU |
|  | Conv2D | 32 | 64 | 3 | 2 | 1 | ReLU |
|  | Linear | 1600 | 128 | - | - | - | - |
| Decoder | Linear | 128 | 1600 | - | - | - | - |
|  | ConvTranspose2D | 64 | 32 | 3 | 2 | 1 | ReLU |
|  | ConvTranspose2D | 32 | 16 | 3 | 2 | 1 | ReLU |
|  | ConvTranspose2D | 16 | 3 | 3 | 2 | 1 | Sigmoid |

For the set autoencoder, we use the default implementation of PISA provided in the author's repository[3]. We train PISA with a latent dimension of 256 with a batch size of 32 for 15000 iterations or until the loss has clearly converged.

Unlike Melting Pot, we do not train an image encoder for VMAS environments as observations are already feature vectors. We train PISA with a latent dimension of 72 with a batch size of 256 for 15000 iterations where the loss has clearly converged.

Since VMAS environments are extremely simple, we find that uniformly randomly sampling from the observation space to generate pre-training data works well. This leads to learning a strong autoencoder where the reconstruction loss is very small during policy training. Hence, we use this method in our final results rather than a uniform random policy. Both lead to task-agnostic communication strategies as they are reward-free.

## F    CHOOSING MARL BENCHMARKS

While other well-known MARL benchmarks exist, we choose not to use these as they either do not require communication to solve (Samvelyan et al., 2019), lack sufficient task variation (Samvelyan et al., 2019; Ellis et al., 2022; Kurach et al., 2020), or are not Dec-MDP/POMDPs (Kurach et al., 2020).

While SMAC (Samvelyan et al., 2019) is commonly used in prior literature, many of its environments can be solved with open-loop policies (i.e. with observations of just the agent ID and time step) (Ellis et al., 2022). As a result, communication is not necessary to solve it. While SMACv2 (Ellis et al., 2022) resolves some of these issues, the objective remains simply to kill all the enemy agents. Consequently, this environment does not have enough task variation to test task-agnostic communication, the main contribution of this paper.

Similarly, GRF (Kurach et al., 2020) is not a suitable benchmark to evaluate communication as it provides full pixel-frame observations, thus giving the entire visual state to each agent and eliminating any need for communication. Moreover, GRF allows control over only a single "active" player at a time. Since only one agent is active at any time step, communication is not applicable in the usual sense, and the environment is neither a Dec-MDP nor Dec-POMDP. Like SMAC, GRF also does not feature task variation as the only objective is to win at the game of football.

Melting Pot (Agapiou et al., 2023) represents similarly challenging tasks. Like SMAC and GRF, it features high-dimensional pixel-based observations and complex objectives. It is a new, but state-of-the-art benchmark. VMAS (Bettini et al., 2022) is also a suitably challenging benchmark. The visualisations in VMAS appear simple, but the dynamics are complex, going beyond kinematics by simulating elastic collisions, rotations, and joints. Thus, while the environments are conceptually basic, VMAS still represents a *realistic* challenge to agents.

## G    COMPUTATION, HARDWARE, AND IMPLEMENTATION DETAILS

We implemented our work with the Ray RLLib library (version 2.1.0 for VMAS and 2.3.1 for Melting Pot) and wrote all our models with the PyTorch framework. Our models and policies were primarily trained on individual NVIDIA A100 GPUs with 40GiB of memory and NVIDIA RTX 2080Ti GPUs with 11GiB of memory. Experiments were conducted with 5 workers for VMAS with 32 vectorised environments and 2 workers for Melting Pot. In each case, we used a single driver GPU while environment simulations were carried out on CPU. Training a policy for 12M environment steps on a VMAS task took approximately 6-12 hours, while 6.4M environment steps on Melting Pot took about 18-24 hours. Pre-training the image encoder took about 6 hours and pre-training PISA took about 1 hour for VMAS and 6 hours for Melting Pot.

---

[3]`https://github.com/Acciorocketships/SetAutoEncoder/tree/main`

