# OpenReview forum: "Generalising Multi-Agent Cooperation through Task-Agnostic Communication"
_ICLR.cc/2024/Conference — Submitted to ICLR 2024_

### Official Review · Reviewer_82Fi · 2023-10-31

**Soundness:** 3 good
**Presentation:** 2 fair
**Contribution:** 2 fair
**Rating:** 5
**Confidence:** 4

**Summary:**

This paper proposes a communication strategy for cooperative multi-agent reinforcement learning that is task-agnostic and environment-specific. The authors pre-train the communication strategy in a self-supervised manner using a set autoencoder, enabling seamless adaptation to new tasks and scaling to more agents. The proposed strategy is compared to task-specific methods and a baseline that does not use communication and is shown to outperform both in different MARL scenarios. The authors also demonstrate that the proposed method can handle variable numbers of agents and can detect out-of-distribution events in the environment. Overall, the paper presents an approach to reducing the inefficiency of task-specific communication strategies in MARL.

**Strengths:**

One strength of this paper is the proposed task-agnostic communication strategy, which is shown to outperform task-specific methods and a baseline that does not use communication in diverse MARL scenarios. The self-supervised and pretrained autoencoder is assumed to capture the representation of a given environment, and each of the policies of agents rollout actions based on the representation. This representation also accelerates policy adaptation to new tasks making the learning more efficient compared to task-specific MARL methods. The proposed framework is also tackling the OOD events in MARL. Code is written satisfyingly, and easy to reproduce.

**Weaknesses:**

The weakness of the approach lies in the method of generating offline datasets, which involves applying a random policy in different environments. In such a setup, the offline dataset often contains episodes with low returns, and with these episodes, it is hard to learn an informative representation due to this inefficient exploration strategy. It is essential to acknowledge that the quality of the dataset is crucial, given that the encoder aims to distill intrinsic information from the environment.

Another potential issue is dataset bias. The authors have not explicitly stated that the dataset was generated across different tasks. In a task-agnostic setting, the learned latent variable should ideally capture data from an environment with a distribution of tasks, facilitating the learning of environment-specific representations.

The adaptability in the task-agnostic setting is not well defined. While it is important to evaluate the task-agnostic setting across various tasks in each environment, the description of how different tasks are generated is not clear. For instance, in the "Melting Pot" collaborative cooking environment, it's unclear whether different tasks involve cooking various recipes or collaborating in different pre-defined ways. Moreover, the shared global reward structure might lead to suboptimal cooperation, with a few agents doing most of the work while others contribute little.

The same for the Discovery, where the differences between tasks appear to be primarily related to the differently generated locations of targets. If I am correct, this might not provide a strong basis for presenting distinct tasks.

The evaluation presented is incomplete. The author only demonstrates results for a single task rather than a distribution of tasks. A robust task-agnostic method should ideally outperform other baselines across a range of tasks within a specific environment.
OOD setting is interesting, but I was wondering if it possible to unfreeze the parameters of the autoencoder when training policy under the OOD setting.This would shed light on how adaptive the communication strategy truly is.

Environments are limited to 2-D environments. Some more complicated 3D environments are truly expected.

**Questions:**

Q1. Could you clarify how you distinguish between tasks within each of the environments?
Q2 Would it be possible to construct an expert offline dataset?
Q3 Could you provide an explanation of what the "no-comms baseline" entails?
Q4 Would it be possible to illustrate the trajectory of the Center of Mass moves in policy training phase, and also compare it to the scenario where the policy is trained with randomized s_t? I believe it gives a more intuitive demonstration how s_t influences learning.

---

> ### Author Response · Authors · 2023-11-15
> **Response to Reviewer 82Fi (1/2)**
>
> Thank you for your review. We’re glad to hear that you found the OOD setting in our work particularly interesting. We would like to address all of your concerns and answer your questions.
>
> > the offline dataset often contains episodes with low returns, and with these episodes, it is hard to learn an informative representation due to this inefficient exploration strategy.
>
> Great point. The offline dataset may contain episodes with low returns with respect to one of the tasks within the environment that we learn online. Fortunately, our autoencoder is trained without the reward. This means that even in very sparse reward scenarios, we should be able to learn a good state representation. If this remains an issue, future work could mitigate this problem (without biasing the offline dataset) by using curiosity-driven exploration (Pathak et al., 2017) instead to sample sparse states more frequently (see our limitations section).
>
> > Another potential issue is dataset bias. The authors have not explicitly stated that the dataset was generated across different tasks. In a task-agnostic setting, the learned latent variable should ideally capture data from an environment with a distribution of tasks, facilitating the learning of environment-specific representations.
>
> Our offline dataset is collected through random exploration in the environment. As a result, there is no reward signal (and hence, no task bias) in our offline data. The task-agnostic strategy we learn can therefore be applied to any task in the environment via defining any reward function within it. For example, when we pre-train the set autoencoder for the Discovery task, we sample data randomly from the environment. This environment can have the Discovery task or the Flocking task (or any other suitable task) defined within it. Since we sample randomly, there is no task bias in the offline dataset.
>
> > the description of how different tasks are generated is not clear.
>
> Please see our comment further below about how tasks are distinguished. Generally, tasks are taken from pre-existing ones in the benchmark that share the same environment.
>
> > the shared global reward structure might lead to suboptimal cooperation
>
> Indeed, you are correct. This is a property of fully cooperative reinforcement learning with Dec-MDPs and Dec-POMDPs. Most fully cooperative RL research studies these decision processes. This remains an open problem and the focus of future research in the field.
>
> > A robust task-agnostic method should ideally outperform other baselines across a range of tasks within a specific environment.
> > Environments are limited to 2-D environments. Some more complicated 3D environments are truly expected.
>
> A limitation of existing benchmarks is that it is very difficult to find a range of challenging alternative tasks from the same environment. We have done our best to accommodate this gap in the field by demonstrating our task-agnostic method is effective in multiple different types of environments, action and observation spaces, and simulators for at least two different tasks in the same environment. While we developed this paper, we deliberated extensively around which environments are suitable to test this work. We have included a discussion around this in Appendix F, including reasons why 3D environments like SMAC and GRF are unsuitable. In addition, we would like to highlight that Melting Pot is becoming a recognised MARL benchmark, featuring as a NeurIPS 2023 challenge [B] and that VMAS has been used in recent state-of-the-art work [C, D], including being directly supported by the new TorchRL library [E]. Both are known and growing benchmarks in the community.
>
> > OOD setting is interesting, but I was wondering if it possible to unfreeze the parameters of the autoencoder when training policy under the OOD setting.This would shed light on how adaptive the communication strategy truly is.
>
> This is an interesting point. However, we believe fine-tuning task-agnostic communication is out-of-scope as it does not fit the objective of our work, which is to achieve sample-efficient communication by training communication only once and deploying it to solve multiple tasks. Thank you for raising this point. It is still interesting to consider ways to improve adaptability without sacrificing task-agnostic communication by, for example, improving unbiased offline dataset collection instead.

---

> > ### Author Response · Authors · 2023-11-15
> > **Response to Reviewer 82Fi (2/2)**
> >
> > > Could you clarify how you distinguish between tasks within each of the environments?
> >
> > A task $\tau$ within an environment $\mathcal{E}$ is defined by a reward function $R_{\tau, \mathcal{E}}(s, a)$ (please see our explanation in the introduction). Thus, two tasks within the same environment are distinguished by their environment and task-relative reward functions. For example, in collaborative cooking, the reward for *cramped* and *circuit* differ due to alternative arrangements of the cooking stations, pots, and ingredients. The rewards yielded for the same agent state are different as a result. Similarly, going from Flocking to Discovery, the reward is now for agents to "cover" targets rather than flocking amongst them.
> >
> > > Would it be possible to construct an expert offline dataset?
> >
> > Great question! In order to be task-agnostic, the offline dataset should be collected without being explicitly biased towards a particular task reward (for us, a random policy was a simple way to achieve this). And so, defining what makes an “expert” is difficult when there is no reward signal to optimise. Perhaps it could be a general policy that is good at all the possible tasks in the environment. If we had such a policy, there would however be no need for pre-training since we can deploy it online. Or more generally, an expert offline policy which is able to explore the sparse states in the environment effectively could be useful—curiosity-driven exploration (Pathak et al., 2017) is one approach we suggest for future work to explore.
> >
> > > Could you provide an explanation of what the “no-comms baseline” entails
> >
> > Thank you for raising this issue. We will update the paper to clarify the difference between no-comms and comms. For posterity, the difference is that the "no-comms" baseline involves training the agents to execute the task in the environment with IPPO (just as we do in the "task-agnostic" and "task-specific" baseline), however, the agents do not share any information (i.e. there is no communication). An agent's policy is conditioned only on its own observation, $o_i$ instead of $o_i$ and $\hat{\mathbf{s}}$.
> >
> > Thanks again for your review and raising so many interesting points. We have included all changes to the paper in orange in the revised PDF. Is there anything else you would like to discuss to increase your score?
> >
> > [B] https://www.aicrowd.com/challenges/meltingpot-challenge-2023
> >
> > [C] Heterogeneous Multi-Robot Reinforcement Learning. AAMAS 2023: 1485-1494
> >
> > [D] https://github.com/facebookresearch/BenchMARL
> >
> > [E] TorchRL: A data-driven decision-making library for PyTorch. CoRR abs/2306.00577 (2023)

---

### Official Review · Reviewer_NAUT · 2023-11-01

**Soundness:** 2 fair
**Presentation:** 4 excellent
**Contribution:** 2 fair
**Rating:** 3
**Confidence:** 3

**Summary:**

The paper focuses on solving multi-agent dec-MDPs. Specifically, the paper proposes a method that uses the local observations of nearby agents to estimate a latent global Markov state. The paper shows that the method can be used to execute unseen tasks within the same environment.

**Strengths:**

The paper is well-written. It is well-organized and the ideas are exposed simply. The problem of multi-agent task execution is clearly important and the idea of having task-agnostic communication should be important for efficiently solving unseen tasks within an environment. The used set auto-encoder that is permutation invariant and can handle variable number of agents is also interesting and promising.

**Weaknesses:**

The problem being addressed is on the execution of multi-agent tasks in unseen environments. In those environments, the agents can communicate their local observations with each other if they are closer than a certain threshold. The method proposed encodes these observations into a latent state. The encoder was learned in a different task than the test.

The paper compares the performance of their task-agnostic method with one where the agent don't communicate at all (no-comms) and one where the encoder was learned on a different task (task-specific). The task-specific method is not explained in the main text (how and what does it learn?), so it is hard to understand how it can be compared with the task-agnostic method. The no-comms baseline is also not expected to be a strong competitor since it has less information than the other too. There is a clear baseline missing, which is a method that receives the observations of the agents that are close but does not use the set autoencoder, but rather a standard neural network architecture. Otherwise, there is no way to say that the set autoencoder, the main contribution within the method, is useful. It is also important to know how many times the agents are communicating. If the agents are always close by each other, than the Markov state is always being observed. Some metrics would be useful.

The method is build in the hope that the set autoencoder is able to predict the Markov state from the local observations. The Markov state is the joint observations of all agents. Therefore, the agents must be able to predict the other agents observations from their own local observations and the ones of their closeby neighbors. However, the other agents observations depends on their policy: if the agents are acting randomly, their observations will be some; if they are acting with a different policy, their observations will be other. Therefore, pre-training the method purely offline with random policies may be troublesome and can lead to serious problems in some environments. It would be important to see how the set autoencoder reacts to fine-tuning during task execution to accomodate for this, also compared with the baselines.

The convergence and regret results can be seen as out-of-scope of the work, or distracting, as the method does not contribute in the reinforcement learning, rather the encoder of the observations. It is typical to have encoders in partially observable environments for reinforcement learning so it is not particularly interesting to analyse the convergence and regret in this work in my opinion. The contributions in 4.2 are also much more aligned with the focus and relevant than the ones in 4.3 and 4.4.

The limitation of acting on a Markov state is not discussed and it should. Specifically, if the estimated Markov state is useful, it encodes the information from the observations of all the agents. Even though the dimension of the latent vector can be fixed, as the number of agents grows, the information scales exponentially. Methods that prune the communication or communicate only what is important for the task do not have this scale limitation. The limitation that multiple agents also need to coordinate (the Markov state is not sufficient) in multi-agent task executions is also not discussed.

The set autoencoder is not explained in detail in section 3.2 and is a core element of the paper.

Minor: the method does not learn to communicate (neither how to "speak" nor how to "listen"), rather learns to encode observations that are available. The title and the exposition may hint the reader otherwise.

**Questions:**

- why is the focus restricted to dec.MDPs, in opposition to dec-POMDPs? Even though in the latter the joint observation is not necessarily the state, it is a more common setting in multi-agent reinforcement learning and the method proposed can be used to estimate the joint observation instead of the state. The generalization would also allow to use more benchmark environments, for example the ones that are more common in the literature (SMAC, MPE, LBF, ...).
- what would be necessary for the method to be used with heterogeneous agents? By heterogeneous, I mean different action and observation spaces.
- in the experiments, do the policies share parameters, or they learn independently?
- in figure 2, $N$ is the number of agents, and should be $n$ or is the number of closeby neighbors?
- in the conclusions, what does it mean that you use "full connectivity between agents"?
- what is meant by "on-policy" throughout the work? It does not appear to be the usual sense in reinforcement learning (Sutton 2018) where the target policy is the same  the behavior policy.

---

> ### Author Response · Authors · 2023-11-15
> **Response to Reviewer NAUT (1/2)**
>
> Thank you for your review. We are pleased to hear you thought the paper was well-written.
>
> > The task-specific method is not explained in the main text (how and what does it learn?)
>
> Thanks for highlighting this. We have now clarified this point in Section 4.2.
>
> > The no-comms baseline is also not expected to be a strong competitor since it has less information than the other too.
>
> Indeed. We included this baseline to verify that communication was being utilised.
>
> > There is a clear baseline missing, which is a method that receives the observations of the agents that are close but does not use the set autoencoder, but rather a standard neural network architecture.
>
> Our primary goal is to demonstrate the effectiveness of task-agnostic communication strategies, comparing task-specific and task-agnostic baselines with identical architectures. While a classical autoencoder could be used for both baselines, it wouldn't alter the experiment's outcome in Section 4.2 as our focus isn't on comparing set autoencoders to standard architectures in this context. However, we acknowledge the importance of highlighting the set autoencoder's advantages. We demonstrate its ability to handle varying numbers of agents and scale beyond the training distribution, as detailed in Section 4.3. This capability is unique to our set autoencoder, unlike standard architectures that cannot encode variable-sized sets into fixed-sized latent states. We do not claim the set autoencoder is superior in experiment 4.2, and for scenarios not requiring variable agent numbers, a standard neural architecture with our task-agnostic approach can be used. We appreciate your feedback and are in the process of clarifying these points in our paper.
>
> > if the agents are acting randomly, their observations will be some; if they are acting with a different policy, their observations will be other. Therefore, pre-training the method purely offline with random policies may be troublesome
>
> Great point. The offline dataset may contain episodes with low returns with respect to one of the tasks within the environment that we learn online. Fortunately, our autoencoder is trained without the reward. This means that even in very sparse reward scenarios, we should be able to learn a good state representation. If this remains an issue, future work could mitigate this problem (without biasing the offline dataset) by using curiosity-driven exploration (Pathak et al., 2017) instead to sample sparse states more frequently (see our limitations section). Fine-tuning task-agnostic communication is not a suitable approach. It does not fit the objective of our work, which is to achieve sample-efficient communication by training communication only once and deploying it to solve multiple tasks. We suggest future work should focus on improving unbiased offline dataset collection instead. Thanks for making this important point. We will add more discussion on this topic to the paper.
>
> > The convergence and regret results can be seen as out-of-scope of the work, or distracting, as the method does not contribute in the reinforcement learning, rather the encoder of the observations.
>
> We agree that 3.1 and 3.2 may break the flow of the paper and this could warrant moving them to an appendix. Thank you for affirming this. Nevertheless, these theorems are novel in the context, showing that our approach of reconstructing the Markov state leads to interesting and important properties that are not general to all encoder approaches. For example, Theorem 3.2 warrants great practicality. When there is an information bottleneck (such as the latent state) or when deploying our approach in realistic scenarios, like where communication connectivity between the entire multi-agent team is not possible, we still know that the regret scales linearly. As a result, the paper would be incomplete without stating Theorem 3.1 and 3.2.
>
> > Even though the dimension of the latent vector can be fixed, as the number of agents grows, the information scales exponentially.
>
> Great observation. Whilst we have demonstrated in 4.3 that we can scale far beyond the training distribution, the limitations of the fixed-size latent state can be seen when we reach 10 agents. As you insightfully suggest, to go beyond this, it is possible to incorporate methods which prune communication on top of our work. This would only affect which elements appear in the set that the set autoencoder encodes. As a result, they can be applied to the communication topology without any modifications to our method.

---

> ### Author Response · Authors · 2023-11-15
> **Response to Reviewer NAUT (2/2)**
>
> > The limitation that multiple agents also need to coordinate (the Markov state is not sufficient) in multi-agent task executions is also not discussed.
>
> We would like to check if we agree on the definition of the multi-agent Markov state. In a Dec-MDP, the literature defines the multi-agent Markov state as uniquely defined by the global observation (Oliehoek & Amato, 2016). Coordination among agents is required to recover the Markov state by sharing individual observations to recover the global observation. This Markov state is, by definition of the Markov property [G, page 465], _sufficient_. As such, multiple agents needing to coordinate is not a limitation of our approach, but a property of Dec-MDPs. Thanks for highlighting this point.
>
> > The set autoencoder is not explained in detail in section 3.2 and is a core element of the paper.
>
> Thank you for highlighting. We have added a full description of the set autoencoder to the paper (see Appendix B) and will move the essential components into Section 3.2.
>
> > why is the focus restricted to dec.MDPs, in opposition to dec-POMDPs?.
>
> This is a great question. We restrict our theoretical focus to Dec-MDPs because the field in general has not solved POMDPs or Dec-POMDPs. Focusing on Dec-MDPs allows us to motivate our method by focusing on reconstructing the Markov state which brings several nice results (particularly Theorem 3.2, which is useful in practice). Nevertheless, we agree that Dec-POMDPs are a more common setting in the literature and have included an additional comment in the paper describing how Theorem 3.2 applies to Dec-POMDPs. We emphasise also that the benchmark tasks we use in both VMAS and Melting Pot _are also Dec-POMDPs_. We have included a discussion in Appendix F regarding our choice of MARL benchmarks. In addition, we would like to highlight that Melting Pot is becoming a recognised MARL benchmark, featuring as a NeurIPS 2023 challenge [B] and that VMAS has been used in recent state-of-the-art work [C, D], including being directly supported by the new TorchRL library [E]. Both are known and growing benchmarks in the community.
>
> > what would be necessary for the method to be used with heterogeneous agents?
>
> Interesting point. For heterogeneous/changing policies at runtime, this is already supported. For agents with different action spaces, no changes are required if we use independent policy networks to support varied action spaces. For different observation spaces, it would be necessary to encode the different types of observations into a shared latent space. Then, the set autoencoder can be trained with variables from this latent space. There are many opportunities for future work to apply our method in interesting ways to different types of heterogeneous agents.
>
> > in the experiments, do the policies share parameters, or they learn independently?
>
> For our experiments on Melting Pot, the policies do not share parameters. For experiments on VMAS, they do. Please see Appendix C for further details. Thank you for highlighting this.
>
> > in figure 2, is the number of agents, and should be or is the number of closeby neighbors?
>
> In Section 4.2, $N$ is the number of agents. In real world deployment however, this should be the number of agents within the communication range of agent $i$.
>
> > in the conclusions, what does it mean that you use “full connectivity between agents”?
>
> We assume agents are within the communication range of all other agents in our experiments. In graph neural network-based communication, this is not required as information can be propagated to out-of-range agents via agents that are in range. In the conclusion, we describe how something similar may be implemented using our method to overcome this full connectivity assumption. We will clarify this in the paper. Thank you.
>
> > what is meant by “on-policy” throughout the work?
>
> We use “on-policy” to describe when we learn a policy to execute the task, using a pre-trained set autoencoder to encode the observations we condition the policy on. This is in contrast to “offline pre-training” which is when we train the set autoencoder with randomly sampled global observations.
>
> We hope this clarifies your questions. Thank you for raising many interesting points! We have put all updates to the paper in orange in the revised PDF. Are there any further points you would like to discuss to increase your score?
>
> [A] SMACv2: An Improved Benchmark for Cooperative Multi-Agent Reinforcement Learning. CoRR abs/2212.07489 (2022)
>
> [B] https://www.aicrowd.com/challenges/meltingpot-challenge-2023
>
> [C] Heterogeneous Multi-Robot Reinforcement Learning. AAMAS 2023: 1485-1494
>
> [D] https://github.com/facebookresearch/BenchMARL
>
> [E] TorchRL: A data-driven decision-making library for PyTorch. CoRR abs/2306.00577 (2023)
>
> [F] https://github.com/Acciorocketships/SetAutoEncoder/tree/main/sae
>
> [G] Sutton, R. S., & Barto, A. G. (2018). Reinforcement learning: An introduction. MIT press.

---

### Official Review · Reviewer_FP8p · 2023-11-01

**Soundness:** 2 fair
**Presentation:** 3 good
**Contribution:** 2 fair
**Rating:** 5
**Confidence:** 4

**Summary:**

This paper introduces a universal communication strategy for cooperative multi-agent reinforcement learning (MARL), applicable across various tasks within a specific environment. The authors employ a set autoencoder for pre-training without relying on task-specific rewards, ensuring policy convergence with latent representations. The approach scales to accommodate additional agents beyond the training setup and effectively identifies out-of-distribution environmental events, with empirical results across diverse MARL scenarios confirming its efficacy.

**Strengths:**

The paper is easy to follow with concise expressions and clear logic.

**Weaknesses:**

1. The experimental environment is overly simplistic; I recommend employing more convincing and commonly used benchmarks such as SMAC and GRF.

2. In essence, the paper models the global environment in advance through the use of an autoencoder. If the global state can be reconstructed from the partial state, it implies that the invisible state can be deduced from the visible state. In this case, there would be no need for communication.

Some typos:

page3 As the the global observation

**Questions:**

Please refer to the weakness section

---

> ### Author Response · Authors · 2023-11-14
> **Response to Reviewer FP8p (1/2)**
>
> Thanks for your review. We're glad to hear that you found the paper easy to follow. Below, we would like to address your concerns.
>
> > The experimental environment is overly simplistic; I recommend employing more convincing and commonly used benchmarks such as SMAC and GRF.
>
> Thank you for this observation. We agree that challenging environments are essential to demonstrate the efficacy of research in MARL. While we were developing this work, we deliberated thoroughly on the experimental benchmark, considering both SMAC and Google Research Football (GRF) in the process. We believe we have good reasons for choosing not to use these (perhaps more well-known) benchmarks, as follows:
>
> While SMAC is commonly used in prior literature, many of its environments can be solved with open-loop policies (i.e. with observations of just the agent ID and time step) [A]. As a result, communication is not necessary to solve it. While SMACv2 resolves some of these issues, the objective remains simply to kill all the enemy agents. Consequently, this environment does not have enough task variation to test task-agnostic communication, the main contribution of this paper.
>
> GRF is indeed challenging. However, it is not a suitable benchmark to evaluate communication as it provides full pixel-frame observations, thus giving the entire visual state to each agent and eliminating any need for communication. Moreover, GRF allows control over only a single "active" player at a time. Since only one agent is active at any time step, communication is not applicable in the usual sense, and the environment is neither a Dec-MDP nor Dec-POMDP. Like SMAC, GRF also does not feature task variation as the only objective is to win at the game of football.
>
> We believe Melting Pot represents similarly (if not more) challenging tasks. Like SMAC and GRF it features high-dimensional pixel-based observations and complex objectives. It is a new, but state-of-the-art benchmark, even featuring as a NeurIPS 2023 challenge [B]. Hence, it is quickly becoming a recognised MARL benchmark. VMAS is also an existing and known benchmark, having been used in recent state-of-the-art work [C, D], including being directly supported by the new TorchRL library [E]. The visualisations of VMAS appear simple, but the dynamics are complex, going beyond kinematics by simulating elastic collisions, rotations, and joints. Thus, while the environments are conceptually basic, VMAS still represents a significant (and _realistic_) challenge to agents.
>
> Finally, we would like to emphasise that although we use IPPO in our experiments, our method is algorithm-agnostic. As new state-of-the-art algorithms and benchmarks appear, our approach can be applied to these new algorithms to solve even more challenging problems as the field evolves.
>
> Thank you for bringing this nuanced issue to light and allowing us to discuss this problem in detail. We have added wording to further elaborate our choice of simulators in the paper to clarify our rationale (in particular, please see Appendix F). We will continue to consider what benchmarks are available and update the paper if we can find suitable environments to conduct further tests on. We believe challenging MARL benchmarks are essential for the progression of the field.
>
> > If the global state can be reconstructed from the partial state, it implies that the invisible state can be deduced from the visible state. In this case, there would be no need for communication.
>
> Interesting point! We are glad to discuss this. Consider a problem, like Discovery, where agents must "cover" targets (i.e. have multiple agents come within some range of a target) to gain reward. Now, if the agents are invisible, the only way to solve this task is for the agents to communicate their positions and whether they have discovered a target in order for other agents to go to them and cover the target. Thus, communication is required.
>
> This example elucidates the more general point that in a Dec-MDP (and in Dec-POMDPs in general), the global state cannot be reconstructed from the partial state seen by an agent. In our work, we do not imply that the global state can be reconstructed from the partial state. We attempt to approximate the global state using the information that we have (whichever observations we have received from other agents). Our set autoencoder has learned the latent information/underlying patterns during pre-training which allow it to attempt to do so. In a Dec-MDP, when we have information from all agents, we have enough information to theoretically reconstruct the global state—but communication is required in order to get information from all other agents.
>
> Thank you for highlighting this important point. We are working to make this clearer in the paper.

---

> ### Author Response · Authors · 2023-11-14
> **Response to Reviewer FD8p (2/2)**
>
> > Some typos: page3 As the the global observation
>
> Great catch! We have fixed this in the paper. Thank you.
>
> Thank you, again, for your review and allowing us to have this detailed discussion. We have included all updates we have made to the paper in orange in the revised PDF. Are there any further points you would like to discuss with us in order to increase your score?
>
> [A] SMACv2: An Improved Benchmark for Cooperative Multi-Agent Reinforcement Learning. CoRR abs/2212.07489 (2022)
>
> [B] https://www.aicrowd.com/challenges/meltingpot-challenge-2023
>
> [C] Heterogeneous Multi-Robot Reinforcement Learning. AAMAS 2023: 1485-1494
>
> [D] https://github.com/facebookresearch/BenchMARL
>
> [E] TorchRL: A data-driven decision-making library for PyTorch. CoRR abs/2306.00577 (2023)

---

### Official Review · Reviewer_CDxU · 2023-11-02

**Soundness:** 3 good
**Presentation:** 4 excellent
**Contribution:** 2 fair
**Rating:** 5
**Confidence:** 4

**Summary:**

This paper decouples "environment" from "task" for MARL by introducing a task-agnostic, environment-specific communication strategy. This addresses the issue of sampling inefficiency and enables adaptation to novel tasks, more agents, and out-of-distribution events without relearning or fine-tuning the communication strategy.

**Strengths:**

In the experiments, the task-agnostic version outperforms others.

**Weaknesses:**

1. Task-agnostic communication for MARL is not a new thing, especially the ones using natural languages (e.g. https://arxiv.org/pdf/2107.09356.pdf). This work did not show comparisons with those methods.

2. Population invariant feature aggregation in MARL is not a new thing, e.g. Evolutionary Population Curriculum（EPC）achieved this using attention mechanisms. Population-invariant communication is also mentioned in papers like https://arxiv.org/pdf/2302.03429.pdf.

3. Theorem 3.1 and 3.2 seem to be trivial.

**Questions:**

1. What are the detailed structures of the autoencoder?

2. What is the exact version of the task-specific algorithm used in experiments?

---

> ### Author Response · Authors · 2023-11-14
> **Response to Reviewer CDxU (1/2)**
>
> Thank you for your review and comments. We are especially appreciative of your efforts to highlight important related work. Below, we would like to address your concerns.
>
> > Task-agnostic communication for MARL is not a new thing, especially the ones using natural languages (e.g. https://arxiv.org/pdf/2107.09356.pdf). This work did not show comparisons with those methods.
>
> Thank you for bringing this to our attention. We have cited the papers you have discussed in our related work section---thanks for highlighting.
>
> The natural language method is particularly interesting. However, we do not believe that it represents a task-agnostic method, and is generally not comparable with our work. Mainly, this is because their communication method is trained with RL in baseline (1), so is not task-agnostic since the communication messages will be biased towards solving one objective. In baseline (2), they do not use RL, but solve a planning problem with communication. Additionally, their method supports only one-way communication, while we send *and* receive messages between all agents. Finally, we note that the authors restrict themselves to using a discrete set of communication symbols (which is not continuously differentiable, as ours is). This means that they do not learn a communication strategy grounded in the environment. Selecting the communication symbols also biases the communication language and hence, in this way also, their approach is not task-agnostic. Furthermore, this limits the generalisation of their method to other tasks as their vocabulary is tailored towards the maze solving task.
>
> Overall, this work is not comparable with ours for the reasons we outlined above, however, it is insightful and we are glad you brought it to our attention.
>
> > Population invariant feature aggregation in MARL is not a new thing, e.g. Evolutionary Population Curriculum（EPC）achieved this using attention mechanisms. Population-invariant communication is also mentioned in papers like https://arxiv.org/pdf/2302.03429.pdf.
>
> Indeed, it is not new. Thank you for highlighting the EPC and SPC paper. It's fascinating work. We have added them to our related work section.
>
> We would like to emphasise that we do not claim supporting variable numbers of agents (population-invariant communication) is novel. Thank you for highlighting this point. We have made the paper clearer on this front (please see the second paragraph of the related work section). There are several works that we have cited in the paper that also support population-invariant communication.
>
> Generally, what we aim to highlight is that population-invariant communication is a _feature_ of our approach (by utilising the set autoencoder). Whilst we could have achieved task-agnostic communication without population-invariance, our method brings this benefit as a bonus. In particular, none of the self-supervised communication methods we discussed in the paper supported population-invariance, so it is an additional benefit that our self-supervised approach _does_ support this in addition to being task-agnostic. Thanks again for making this point more salient. We have now clarified this in the paper.
>
> > Theorem 3.1 and 3.2 seem to be trivial.
>
> 3.1 and 3.2 are simple and we agree that this may warrant moving them to an appendix. Thank you for affirming this. Nevertheless, these theorems are novel in the context, showing that our approach of reconstructing the Markov state leads to interesting and important properties. For example, Theorem 3.2 warrants great practicality. When there is an information bottleneck (such as the latent state) or when deploying our approach in realistic scenarios, like where communication connectivity between the entire multi-agent team is not possible, we still know that the regret scales linearly. As a result, the paper would be incomplete without stating Theorem 3.1 and 3.2. We agree that it should perhaps consume less of the main body and we appreciate you highlighting this.

---

> ### Author Response · Authors · 2023-11-14
> **Response to Reviewer CDxU (2/2)**
>
> > What are the detailed structures of the autoencoder?
>
> Thank you for highlighting that this is missing. We are working towards ensuring that the paper is completely self-contained. We have now added a full description of the set autoencoder to the paper (see Appendix B).
>
> A description of the architecture of the autoencoder can also be found in (Kortvelesy et al., 2023, Figure 1). In this architecture, an encoder assigns input elements (known as "keys") to values and encodes them with a neural network. Simultaneously, it encodes the keys with another network. The sum of the element-wise products of the encoded values and keys forms the latent state to which a cardinality embedding is added. A similar decoding process decodes the cardinality of the set and produces the corresponding number of keys and "queries". The element-wise product of the queries by the keys produces a "disentangled" encoding which is decoded by a final network that produces the reconstructed set. For details of the hyperparameters we use for this set autoencoder, please see Appendix E (particularly paragraph 2). Note that we use the default implementation provided by the author, which can also be found at [A].
>
> Thank you again for pointing out this gap in the paper. We have added a more complete description of the architecture.
>
> > What is the exact version of the task-specific algorithm used in experiments?
>
> Great question! We use IPPO, just as we do for the other baselines (we have detailed our training setup in Section 4.1 and Appendix E). The only difference between the task-specific and the task-agnostic baselines is that instead of pre-training the set autoencoder with random samples in the environment, we pre-train the set autoencoder by learning a policy (again with IPPO) to execute a _similar_ (but clearly distinct) task in the same environment. Then, we take this set autoencoder, freeze its weights, and deploy it to learn the main task (just as we do with the task-agnostic baseline). We describe which tasks we do pre-training on for each corresponding main task in Section 4.2. Thanks again for pointing this out. We will ensure to make this clearer in the main text.
>
> We appreciate your interesting questions and insights. We have highlighted any modifications we have made to the revised paper in orange. Please let us know if you have any further questions. Is there anything more we can do to increase your score?
>
> [A] https://github.com/Acciorocketships/SetAutoEncoder/tree/main

---

### Author Response · Authors · 2023-11-18
**Overall comment (+ summary of changes)**

We would like to thank the reviewers for their reviews, the many insightful points that they have raised, and giving us the opportunity to have a nuanced discussion.

We have included changes to the paper in orange in the revised PDF, including:
- A complete description of the set autoencoder (Appendix B)
- Details on Theorem 3.2 applied to Dec-POMDPs (page 4)
- A more complete description of algorithms and baselines (page 6)
- Additional related work (page 9)
- Discussion on MARL benchmark choice (Appendix F)

In general, there are few things that we would like to emphasise:

We stress that the core contribution of our paper is the introduction of a task-agnostic communication method for MARL. This is novel, being continuously differentiable and grounded in the environment, in addition to being supported by Theorem 3.1 and 3.2 in the context of Dec-MDPs.

While it would potentially be interesting to study fine-tuning our approach during task learning/execution (in the OOD context or otherwise), this is out-of-scope as our objective is to achieve sample-efficient communication by training communication only once and deploying it to solve multiple tasks.

We also agree that it would be valuable to study more benchmarks. However, we deliberated thoroughly on this during the development of this paper and after an exhaustive search, converged on VMAS and Melting Pot because they are the only benchmarks we are aware of which support sufficiently varied tasks to study task-agnostic communication. Other suggested benchmarks also have issues which prevent them from being useful to evaluate our work (see Appendix F). We will continue to consider what benchmarks are available and update the paper if we can find suitable environments to conduct further tests on. We believe challenging MARL benchmarks are essential for the progression of the field.

Finally, we want to say that we are delighted to see so much interest in our work! We would be glad to discuss further if there are any more questions.

---

### Meta-Review · Area_Chair_E1ti · 2023-12-15

**Metareview:**

This paper proposes an autoencoder-based solution to improve multi-agent cooperation. Authors claim that the proposed solution is task-agnostic and hence better than task-specific communication methods.

Reviewers raised several questions about the work and the authors addressed most of the questions. Even though reviewers did not respond, I read the rebuttal and most of them are addressing the concerns. However, I think this paper needs more work before being ready for publication.

There are 2 major concerns from the reviewers which are very valid:

1. Authors use a random policy to pre-train the autoencoder. This is an issue since if the agents act randomly, their behaviour could be different. The authors claim that future work could address this issue. However, given the simplicity of the proposed solution (some reviewers complaint about lack of novelty but it is ok) I expect the authors to address this issue.

2. The baselines in the paper are not convincing. There is no other task-agnostic baseline considered. Authors should attempt to design simple baselines and show that their auto-encoder based solution is better than them.

**Justification For Why Not Higher Score:**

There are 2 major issues with the paper as mentioned above. The work is not very novel. So at least the experiments have to be very thorough.

**Justification For Why Not Lower Score:**

N/A

---

### Decision · Program_Chairs · 2024-01-16

Reject